# Emptying the Ocean with a Spoon: Should We Edit Models?

**Yuval Pinter** and **Michael Elhadad**
Department of Computer Science
Ben-Gurion University of the Negev
Be'er Sheva, Israel
{uvp,elhadad}@cs.bgu.ac.il

## Abstract

We call into question the recently popularized method of direct model editing as a means of correcting factual errors in LLM generations. We contrast *model editing* with three similar but distinct approaches that pursue better defined objectives: (1) *retrieval-based architectures*, which decouple factual memory from inference and linguistic capabilities embodied in LLMs; (2) *concept erasure methods*, which aim at preventing systemic bias in generated text; and (3) *attribution methods*, which aim at grounding generations into identified textual sources.

We argue that direct model editing cannot be trusted as a systematic remedy for the disadvantages inherent to LLMs, and while it has proven potential in improving model explainability, it opens risks by reinforcing the notion that models can be trusted for factuality. We call for cautious promotion and application of model editing as part of the LLM deployment process, and for responsibly limiting the use cases of LLMs to those not relying on editing as a critical component.

## 1 Introduction

Large language models, or LLMs, have taken the NLP world by storm. After originally focusing on training them as a vehicle for transfer learning, recent advancements have cast LLMs in the role of one-stop shop, all-knowing oracles. One particular finding that has contributed to this reframed usage is the apparent **factuality** properties of pre-trained LLMs: somehow, mere next-word prediction training has produced models that can complete certain correct facts about the world when asked to do so. LLMs have been perceived by the public at large as a replacement for search engines, and the scant disclaimers appearing in commercial services offering LLM querying do not appear to have made a dent in this perception.

Despite these developments, LLMs of course keep making mistakes. There is, after all, an inherent mismatch between their pre-training objective and the desire for factuality. In recent years, researchers have come up with several remedies to the problem of nonfactual LLM outputs, one of which being **model editing**, where parameters inside an LLM are tweaked based on individual facts marked for correction. These works focus on solving problems such as ensuring stability of other factual outputs following the editing, or batch editing, or computationally efficient editing. In this opinion paper, we call into question the entire *idea* of model editing, citing concerns about intended use cases, conceptual scalability, potential for bias, safety, and overall accountability. We advocate using models that have explicit knowledge modules for tasks requiring factual knowledge, and other existing techniques to mitigate the need for model editing in the first place. Having said that, we acknowledge the usefulness of direct editing for certain use cases such as interpretability probes, and recommend limiting editing to such scenarios.

## 2 Model Editing

Sinitsin et al. (2020) first introduce the notion of updating large ML models to account for local performance expectations motivated externally. They cite cases where mistakes are critical, like object detection in self-driving cars. Later work suggests model editing can aid in protecting privacy and eliminating bias (Zhu et al., 2020), and as a measure of "catching up" with time-sensitive facts such as "PM of the UK" changing over time (Bommasani et al., 2021; Mitchell et al., 2022), or as a means of understanding the mechanics of black box models (Meng et al., 2022). Sinitsin et al. (2020) specify several desired properties of editing methods: **reliability** (the target facts are updated as intended), **locality** (other changes don't happen; a measure for this property is termed **drawdown**), and **efficiency** (successful edits are computationally light); later work (De Cao et al., 2021) added **generality** (the

ability to modify models not originally trained for knowledge retention), **consistency** (robustness to paraphrasing, in the specific use case of text models), and an unspecified property we'll call **frugality** (only minimal components of the model are changed during editing). A well-studied limitation, which most of these works focus on mitigating, and related to locality, is catastrophic forgetting (Ratcliff, 1990), i.e., ensuring that an edited model does not lose performance on tasks for which it was explicitly trained and performs well on.

Methods for model editing have evolved from editable training (Sinitsin et al., 2020), a procedure requiring an a-priori decision that a model would later be edited, to the locality-motivated changing of specific parameters within models (Zhu et al., 2020; Meng et al., 2022). Recent work (Mitchell et al., 2022) draws attention to the danger of model performance degradation accumulating over the course of many successive edits, and seeks mitigation through improved methods. Hase et al. (2023b) extend the consistency requirement to hold over entailed and equivalent facts as well as paraphrases, suggesting model editing as a way of reconciling cases where certain facts are produced correctly but those entailed from them are not.

## 3 Critique

In this section, we present the case against model editing as a practice, regardless of the performance obtained by any single method. We begin with an analysis of the premise underlying the model editing research objective: the hypothesis that LLMs can be used as fact repositories. We then focus on reasons for which editing facts cannot be designed a-priori as a means to maintaining fact-providing LLMs, and continue with practical considerations for why even this ill-posed goal is probably unattainable.

### 3.1 LLMs as Knowledge Repositories?

The perception that LLMs can behave as knowledge repositories was first formulated and experimentally supported with the introduction of the LAMA benchmark (Petroni et al., 2019), where pretrained language models were queried in a zero-shot setting against 51K knowledge triplets extracted from knowledge banks and reformulated as fill-in-the-blank statements. 2019-level models (BERT-XL) got the top answer correctly about 26.5% of the time. The limitations of the LAMA

study were later studied and this result was shown to not be robust against multi-token spans (as opposed to a single token answer; Kalo and Fichtel, 2022), in an open vocabulary setting (as opposed to a closed vocabulary of 21K tokens; Roberts et al., 2020), and when queries vary (Kassner and Schütze, 2020); indicating that the LAMA experiment relied on heuristics to predict answers. Since then, more realistic benchmarks have been introduced, as well as more robust querying techniques (Jiang et al., 2020), to address some of these limitations. As the scale of LLMs increased, recent work has also scaled up the size of benchmarks, shifting the focus to facts concerning rare entities, where even recent LLMs struggle (Kandpal et al., 2023; Sun et al., 2023). These experiments indicate that an LM's ability to answer a question depends on the number of times information relevant to that question appears in the pretraining data. As a result, LLMs cannot be safely used to answer queries about the long-tail of facts that have been rarely mentioned in pretraining (Mallen et al., 2023).

Beyond the capability to reliably answer queries, LLMs should also fulfill other requirements in order to be considered as fact repositories (AlKhamissi et al., 2022): the ability to edit knowledge (add, delete, update facts), logical consistency (answers to different but related facts must agree), reasoning (the ability to infer additional answers on the basis of logical rules), and explainability and interpretability (supporting an answer through a convincing chain of arguments). Experimental results assessing these dimensions indicate that current LLMs fail on all of them. He et al. (2023) demonstrate that LLMs underperform on computing ontology subsumption inference when compared with other trained approaches such as NLI models and symbolic systems such as OWL Reasoners (Glimm et al., 2014).

The evidence supporting the premise that LLMs can be used as fact repositories is, thus, currently weak. Beyond this fragile foundation, we focus below on specific arguments regarding the capability to edit models as if they were fact repositories.

### 3.2 Systemic Mismatch

A very basic property of LLMs which contrasts with their usage as knowledge repositories is their *stochastic* nature. When used for augmenting creative endeavours, for data exploration, or for free-form tasks such as summarization, this property

is desirable: the use case calls for variation or unexpectedness. In other cases, we may be content with models that supply us with a distribution of outputs, from which we can estimate the probability of individual responses and adjust our expectation for a reliable output. Since the latter is absent from many models available only through 3rd-party APIs, we are left with obtaining text generated from an unknown distribution, which we argue is insufficient for factuality-dependent applications.[1] It can even be argued that getting facts wrong is a *feature* of vanilla LLMs rather than a bug: as their core training procedure aims to simulate plausible continuation of text, it should not surprise us that models would repeat widely-assumed falsehoods in a way that negates the idea of using them for factual purposes. If most people think, and write, that Los Angeles is the capital of California,[2] an LLM is *supposed* to complete a relevant prompt accordingly. There is also no built-in reliability or robustness in LLMs that sample outputs from a distribution: two instances of the same prompt can easily produce contradicting facts, and indeed often do.

Additionally, the idea of editing facts in a model suggests that we always *want* a model to supply us with a fact as an answer to a question. However, at times, questions may be posed while pre-supposing or otherwise assuming harmful propositions such as stereotypes or conspiracy theories. Editing the "fact" associated with the question "which government agency faked the moon landing?" would not provide us with an improved model; what we may want is to *remove* such facts altogether, or provide the model with a means of challenging the presupposition, or avoiding giving any answer at all. At the same time, many relations that we would term "facts" can be argued to be vital notions without which certain kinds of basic communication is impossible.[3] An LLM that cannot assert whether trees have leaves, or asserts that they never do, is in danger of becoming irrelevant for most tasks requiring any kind of interaction with the world. As philosophy and practice surrounding these questions progresses, we can hope this gap between 'must-know' and 'must-not-know' will eventually tighten to form workable bounds on LLM knowledge capacity.

## 3.3 Architectural Implausibility

Estimates vary wildly, but there are over 100 million notable facts in the world.[4] Even the cut-off for what constitutes a fact is unclear. Does a .3% change in a demographics statistic, or a new esoteric sports record, call for an edit? Do the daily whereabouts of world leaders constitute facts? What about those of celebrities or journalists? As events unfold daily in world politics, economics, sports, and many other walks of life, facts are added and changed in larger quantities and rates than can be plausibly "caught up with" through surgical model editing, akin to emptying the ocean with a spoon.[5] If we choose to limit ourselves in which facts we find important enough to edit, we introduce bias into the system, opening the door to a host of documented harms which permeate many language technologies (Chang et al., 2019; Blodgett et al., 2020). This choice can be implicit as well as explicit, and is incredibly hard to avoid.

In a similar vein, the vastness and variability of facts is likely to lead to bias in evaluating the complement set of edits, those facts controlled for as *not* changing following an edit (drawdown). Even paraphrases of edited facts are not guaranteed to change alongside the selected phrasing (De Cao et al., 2021), nor are entailed facts (Hase et al., 2023b). This problem also manifests as a safety issue, since unchecked facts can turn out to be quite important for model usage, but perhaps taken for granted (or simply not explicitly covered) when designing a drawdown benchmark.

There is evidence (Mallen et al., 2023; Jang et al., 2021) that facts above a certain "popularity threshold", measured in terms of views on Wikipedia articles, are harder to edit out of models compared to facts lying on the long tail of view distributions. Inherently being out of sight, the unpopular facts thus become susceptible to the double risk of being edited alongside target facts while not being deemed important enough to be checked during drawdown testing. The ultimate outcome of such procedures can be a monolithization of LLM-supplied "knowledge" that focuses on certain popular domains and interests while losing all usefulness in many topics contributing to the extensive diversity of the human and natural experience.

---

[1]Called "transparent-sensitive tasks" (Luo et al., 2023).

[2]The capital of California is Sacramento.

[3]We thank a reviewer for making this point.

[4]107,323,022 appear in Wikidata as of October 17, 2023, conforming to the definition of *notable items* used in https://www.wikidata.org/wiki/Help:Items.

[5]Attributed, according to the Internet, as a "Yiddish proverb", but, hey, who knows?

Empirical evidence indicates existing editing models fail to properly account for the ripple effect of a fact editing operation (Cohen et al., 2023). For example, *the insertion of the fact "Jack Depp is the son of Johnny Depp" introduces a "ripple effect" in the form of additional facts that the model needs to update (e.g. "Jack Depp is the sibling of Lily-Rose Depp")*. Results in symbolic approaches to this task have demonstrated that this knowledge updating task is of high computational complexity, even NP-hard, for example in Truth Maintenance Systems (TMS; Rutenburg, 1991). These results carry to approaches based on machine learning techniques (Knoblauch et al., 2020). There is, therefore, theoretical basis to conclude that model editing will at best address the problem of consistent updating in a roughly approximate manner, and most likely fail to update rarely seen facts within the ripple effect of editing operations.

Finally, recent empirical findings show additional weaknesses of edit methods by extending the evaluation suites to cover aspects beyond fact editing metrics, such as specificity and robustness of the models post-editing (Onoe et al., 2023; Hoelscher-Obermaier et al., 2023; Hase et al., 2023a; Brown et al., 2023).

## 4  Alternatives to Model Editing

Model editing is motivated by the desire to control the text generated by LLMs to make them more compliant with desirable outcomes, specifically to control factuality: when an LLM generates text expressing an incorrect or obsolete fact, remove the fact from the LLM's "memory" and try again (Peng et al., 2023). Other relatively direct approaches to improve LLM factuality (and, in general, to control and revise the text generated by LLMs) have been proposed, most notably Reinforcement Learning with Human Feedback (RLHF; Bai et al., 2022; Ouyang et al., 2022; Ramamurthy et al., 2023; Rafailov et al., 2023; Carta et al., 2023), or adding a factuality objective to the training procedure (Lee et al., 2022). We now survey approaches which define a more achievable objective than that pursued by direct model editing or output manipulation. These approaches avoid the assumption that the LLM is a fact repository, and thus steer clear of the attempt to update this repository in a logically consistent manner, which is computationally hard to achieve and verify. While these alternatives avoid the more problematic aspects of model editing, they still suffer from their own limitations, but we argue that they identify more promising research directions.

**Incorporating Knowledge Bases**  In *retrieval-based models*, factual knowledge is explicitly represented in a dedicated component external to the LLM. The way this external fact store is represented and combined with the LLM varies: it can be a collection of textual documents that is searched using a text retrieval component, or an RDF graph, or encoded as a set of vector embeddings, or represented as modular expert LMs trained on curated datasets. In all cases, in the retrieval-based approach, the model can explicitly cite the source which underlies a specific generation, and let the user decide its credibility.

Once external (non-parametric) knowledge is retrieved, it must be composed with the LLM generation process. Khandelwal et al. (2020) introduce a k-nearest neighbor method with interpolation; RETRO (Borgeaud et al., 2021) combines the prompt with retrieved documents through a specialized cross-attention mechanism; other LLM-IR variants include a fusion-in-decoder method (ATLAS; Izacard et al., 2022) and TRIME (Zhong et al., 2022), all retrieval-based models that maintain the capacity for few-shot in-context learning.

Recent work addresses the integration of LLMs and IR by learning to combine results from search engines into the context provided to the LLM to answer a specific question (Xu et al., 2023). SAIL (Luo et al., 2023) introduces an instruction fine-tuning method to ground language generation on search results and learn to select trustworthy sources. CooK (Feng et al., 2023) approaches the task of combining multiple curated modular knowledge sources into an integrative system, where sources are modeled as independent LMs and an integrator LLM combines the information from these modules to answer a question.

In all of these approaches, factual knowledge is stored outside of the parameters of the LLM and can be manipulated without retraining the LLM. These approaches have been shown to be scalable in the number of facts. Editing the fact store means the same as updating a database, thus simplifying the formulation of the task.

Retrieval-based models, arguably, do not resolve all of the concerns we identify with model editing. The problem of identifying the provenance of a given generation span in these combined models

remains acute: the text can be determined by the internal LLM parameters, by the external stores, or by their combination, even if they are not logically consistent with one another. Facts that have been seen more often during LLM training may have more weight in this interpolation even if they are wrong or when they become obsolete. Zhu et al. (2020) claim that "modifying knowledge in explicit memory module networks like FaE (Verga et al., 2020) is not easier and demands changes in the LLM component as well." This effect was also identified in the RealQA baseline test (Kasai et al., 2022): in this benchmark containing questions about time-sensitive data, experiments showed that in many cases GPT-3 (Brown et al., 2020) properly integrated data retrieved from a KB and injected into the context, but often still generated text based on outdated data from the LLM. While the clear separation of the components and the identification of the composition of external knowledge improve transparency-sensitive tasks, the objective of identifying provenance in the generated text and controlling which sources are appropriate in a given context remains an open research question. The formulation of the task of *attributing generated text to identifiable sources* (AIS; Rashkin et al., 2022) is a key contribution to this direction. Neeman et al. (2022) also address an aspect of this issue with a counterfactual data augmentation technique to disentangle contextual and LLM knowledge when generating text.

**Continual training**   focuses on incrementally training a model by introducing new tasks or new domains (e.g., Razdaibiedina et al., 2023). Model editing does not directly fall within this objective, as it concerns updating precise elements in the model while keeping tasks and domains unchanged. Yet, the identification of drawdown in model editing is similar to the risk of *catastrophic forgetting* identified in continual learning. Within this approach, we could situate model editing as a type of re-training or post-training. Zhu et al. (2020) note that just fine-tuning over a set of facts-to-update costs in degradation on other facts. Jang et al. (2021) identify the problem and propose to apply techniques from continual training to the task of incremental updating of LLM knowledge. Overall, while continuous training may seem to *a priori* avoid the risks of the model editing approach, it seems to suffer from many of the main evaluation problems.

**Concept Erasure**   The goal of concept erasure (Elazar and Goldberg, 2018; Ravfogel et al., 2020; Belrose et al., 2023) is to remove unwanted bias from embeddings generated by LLMs and subsequently from the generated text. This goal is motivated by fairness objectives: preventing protected attributes from causally affecting text generation. This motivation is somehow related to that of model editing (prevent damage from generated text) but key differences exist: (1) the method addresses general, well-identified concepts (protected attributes such as gender, race, age) as opposed to less well-defined specific *facts*; (2) it operates as a post-hoc transformation of embeddings, as opposed to a modification of the model itself, and as such it allows for more accountability than an ever-changing model; (3) it is verifiable with well-defined metrics over a limited pre-declared scope. While concept erasure has more limited scope than model editing, it defines an objective that can be evaluated with aggregated metrics in a robust manner. One possible avenue of future research can be to examine whether the erasure approach can be extended to address specific aspects of factuality, such as temporal validity.

**Maybe it's better to have a model that knows what it doesn't know?**   As identified in Kasai et al. (2022), a prerequisite to avoid generating wrong text is to identify what is not known: "can an open-domain QA system identify unanswerable cases?" The related issue of unanswerable questions has been addressed in a robust way (Rajpurkar et al., 2018; Sulem et al., 2021, 2022); yet the challenge in the context of LLMs and factuality is that problematic questions like those specified in §3.2 do *look* answerable.

## 5   Conclusion

We agree that model editing is an attractive task to define, with clear benchmarks and expected outcomes. However, in current practice it contributes towards unrealistic expectations that we can solve problems like LLM hallucinations, which would lead to potential harm in unleashing use cases that are not in fact within the capabilities of LLMs alone. We advocate for the usage of retrieval-augmented methods and other structural and post-hoc methods in order to achieve the stated large-scale goals, while conceding the benefits of editing to "safer" applications such as model interpretability and robustness checking.

## Acknowledgments

We thank Sarah Wiegreffe for comments on earlier drafts. We thank the reviewers for their valuable feedback. Yuval Pinter was supported in part by the Israeli Ministry of Innovation, Science and Technology (Grant 2022/5451).

## Limitations

This is an opinion paper. We have purposely not made empirical analyses in order to support our critique, knowing that data is contestable and may vary according to its collection methodology. We aim to convince on the merit of examples and rhetorical argumentation rather than concrete evidence, mostly out of our reach for the models under consideration, which we nevertheless urge readers and practitioners to seek and use in attempts to either support or disprove our claims.

## Ethical Considerations

As we were finalizing the publication version of this paper, the Hamas organization launched a murderous attack very close to our home on October 7, 2023. This attack has killed over 1,000 civilians in horrendous manners, including many members of our university and their families, some of our students have been kidnapped and we are still without news about their status. We are still personally under constant attack by rockets launched by Hamas and Hezbollah towards civilian areas (nearly 8,000 rockets have been launched in the past two weeks). Hundreds of thousands of Israeli civilians have been evacuated from their homes to ensure their safety. Palestinian civilians in Gaza are also victims of this war. This situation has prevented us from fully engaging in our research and polishing this paper. Our scientific domain does not provide us with ways to clarify the moral dilemmas that war triggers. The situation in our region stirs strong emotions worldwide, and the media is abuzz with analyses, often lacking depth, knowledge or simple factuality. Given the inability of domain experts to bring clarity when it is so badly needed, we call for humility in technical analysis regarding factuality in LLMs. We do not know what factuality is. However, words and lies harm, and the hatred they distill kills.

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
