# OpenReview forum: "Emptying the Ocean with a Spoon: Should We Edit Models?"
_EMNLP/2023/Conference — EMNLP 2023 Findings_

### Official Review · Reviewer_HGZK · 2023-07-25

**Soundness:** 2

**Excitement:**

2: Mediocre: This paper makes marginal contributions (vs non-contemporaneous work), so I would rather not see it in the conference.

**Paper Topic And Main Contributions:**

The authors call into question the idea that attempts should be made to edit models in order to satisfy factuality requirements, and propose that we rather use dedicated knowledge modules and other post-hoc techniques. They go on to explain their rationale and summarize some recent works that explore alternative strategies.

**Questions For The Authors:**

What do you believe are the main contributions and points of novelty in this paper?

Doesn't the fact that so many recent works are investigating the category of methods you are promoting not indicate that progress is already being made in a direction that would alleviate your main concerns?

**Reasons To Accept:**

The paper is well written and clearly outlines concerns surrounding the model editing methods in question. The authors provide a concise summary of recent works that have developed both model editing and more post-hoc/retrieval based techniques.

**Reasons To Reject:**

While much of the authors' concerns re model editing seem reasonable, it is unclear what exactly they have identified or are proposing that is novel. Across the papers that they cite, the main concerns raised by the authors seem to already be recognised and possible solutions are being investigated. e.g. Mitchell et al., 2022a highlight the problems arising after many successive edits and then actually propose a potential solution that uses an external system for storing content edits and reason over them when needed (of the category of systems championed by the authors).

The moon landing example given on lns 146-157 seems a bit contrived. Prompting ChatGPT, for example, with the question they pose does in fact produce the answer they desire. Also, much of the motivation for fact-oriented model editing is to account for clearly defined instances with a known/desired answer.

The fact that so many alternative methods exist and are referred to makes it somewhat difficult to see what the need for this paper is, when the issues proposed are clearly already being addressed in the ways suggested.

Even though the authors acknowledge it in the limitations section, the fact that this paper also contains no empirical results or examples illustrating the issues discussed doesn't do much to strengthen their argument. If concrete examples of problems resulting from model editing, or some data illustrating the extent to which it has been adopted under the assumption of blanket reliability, were included it would be easier to accept the gravity of their concerns.

**Reproducibility:**

N/A: Doesn't apply, since the paper does not include empirical results.

**Reviewer Confidence:**

4: Quite sure. I tried to check the important points carefully. It's unlikely, though conceivable, that I missed something that should affect my ratings.

---

> ### Author Rebuttal · Authors · 2023-08-24
>
> ## Answers to Questions For The Authors
>
> **Q1: What do you believe are the main contributions and points of novelty in this paper?**
>
> 1. We do not claim for novelty - this is a position paper. We aim to critique a general research approach around LLMs which we identify as ill-defined and risky and aim to raise awareness around the associated risks.
> 2. The hope the conclusion (364-376) summarizes our main contribution:
> > We propose that model editing is an attractive task to define, with clear benchmarks and expected outcomes, however in current practice it contributes towards unrealistic expectations that we can solve problems like LLM hallucinations, which would lead to potential harm in unleashing use cases that are not in fact within capabilities of LLMs alone.
> > We advocate for the usage of retrieval-augmented methods and other structural and post-hoc methods in order to achieve the stated large-scale goals, while conceding the benefits of editing to “safer” applications such as model interpretability and robustness checking.
>
> **Q2: Doesn't the fact that so many recent works are investigating the category of methods you are promoting not indicate that progress is already being made in a direction that would alleviate your main concerns?**
>
> We do not deny the fact that researchers active within the "model editing" approach identify and solve technical issues.
> We instead argue that the underlying assumption that LLMs are fact repositories that can be edited is problematic and leads to specific risks that we aim to identify.
>
> ## Reaction to "Reasons to Reject"
>
> > Across the papers that they cite, the main concerns raised by the authors seem to already be recognised and possible solutions are being investigated. e.g. Mitchell et al., 2022a highlight the problems arising after many successive edits and then actually propose a potential solution that uses an external system for storing content edits and reason over them when needed (of the category of systems championed by the authors).
>
> See response to Q2.
>
> > The moon landing example given on lns 146-157 seems a bit contrived. Prompting ChatGPT, for example, with the question they pose does in fact produce the answer they desire. Also, much of the motivation for fact-oriented model editing is to account for clearly defined instances with a known/desired answer.
>
> The way ChatGPT produces the desired answer is not through model editing - it is most likely through RLHF post-processing (although who knows how proprietary products work). One should not lump together all approaches aimed at avoiding offensive or harmful or untrue LLM output as "model editing" - these are different strategies (see our intro to Section 4 213-225).
>
> The point we make is that the identification of "clearly defined instances" is problematic (it assumes one can enumerate all the *facts* to be processed or rank facts by importance - points we discuss in 3.2). Facts can be introduced in prompts as presupposition or in counter-factual constructs or implied and still lead to specific risks.
>
> > Even though the authors acknowledge it in the limitations section, the fact that this paper also contains no empirical results or examples illustrating the issues discussed doesn't do much to strengthen their argument. If concrete examples of problems resulting from model editing, or some data illustrating the extent to which it has been adopted under the assumption of blanket reliability, were included it would be easier to accept the gravity of their concerns.
>
> We believe that the risks we identify are specific - for example the three arguments in 178-189, 190-204, 205-211 are all supported with specific sources.

---

### Official Review · Reviewer_UmZH · 2023-08-02

**Soundness:** 2

**Excitement:**

2: Mediocre: This paper makes marginal contributions (vs non-contemporaneous work), so I would rather not see it in the conference.

**Paper Topic And Main Contributions:**

This is a position paper arguing against LLM model editing approaches. The particular target of criticism is the class of methods in this space that seek to directly edit model parameters so that the model's behavior reflects a new set of facts (and their consequences). The central criticisms of these approaches offered in the paper is that LLMs are trained with an objective that doesn't support factuality, and that editing approaches won't scale to the number of facts that need to be changed. The paper describes alternative approaches to the core problem of ensuring that our systems produce factual, up-to-date responses.


**Reasons To Accept:**

This is a spirited discussion on a timely topic, and I appreciate the forthright tone that the paper takes when assessing both the editing techniques and the alternatives that the authors prefer (which have major shortcoming as well).


**Reasons To Reject:**

I should say that I share the authors' overall view. I too think that model editing is the wrong bet to make, for numerous reasons. I favor the retrieval-augmented approaches described in section 4.

That said, I feel that the current paper misses the mark. The argument needs to be substantially sharpened in order to have a productive impact:

1. The stated goal of the paper is to "call into question the entire _idea_ of model editing". This is incredibly ambitious. I would not recommend such a sweeping scope for such a short paper. This would require some kind of elegant argument that these approaches cannot, even in principle, serve positive goals. The paper does not offer such an argument.

2. The paper blurs together inherent limitations of LLMs with limitations pertaining to model editing. In particular, the argument in Section 3.1 is not even really about model editing. It's about LLMs and/or LLM APIs. For Section 3.2, the same argument could be used to argue against LLMs -- and even against the whole idea of indexing the Web and using it to learn things about the world!

3. Search engines are vexing for the paper's arguments concerning "factuality". All the concerns raised about factuality hold for traditional search, including the arguments in 3.1. If lots of people on the Web say that Los Angeles is the capital of California, then n-gram-based search will also surface those results. The issue here is conflating factuality with grounding in a database. AI people should never promise factuality, but they should be able to deliver grounding or provenance.

4. The argument in 3.2 (and the metaphor of "emptying the ocean with a spoon") needs to be further substantiated. After all, changing a single node in a knowledge graph can have wide-ranging effects that spread automatically through the network. For example, changing the name of the U.S. president could change the name of the dog owned by the president's mother's first cousin, assuming the knowledge graph has the right edges and so forth. If we could train LLMs to have this kind of structure, then we would be able to change a lot of things with a small database of facts. But, anyway, all of this needs to be properly benchmarked against what we have now with Web search, Wikipedia editing, etc.

5. I appreciate that section 4, on alternative approaches, notes that these alternatives suffer from all of the major problems that the paper identifies with model editing. But this is also a signal that the paper has set its sights too high.

I would encourage the authors to find a more focused sort of criticism and rewrite the paper accordingly. As I said above, I am on their side!

**Reproducibility:**

N/A: Doesn't apply, since the paper does not include empirical results.

**Reviewer Confidence:**

4: Quite sure. I tried to check the important points carefully. It's unlikely, though conceivable, that I missed something that should affect my ratings.

---

> ### Author Rebuttal · Authors · 2023-08-24
>
> ### Reaction to "Reasons to Reject"
>
> > R1: The paper blurs together inherent limitations of LLMs with limitations pertaining to model editing.
> > In particular, the argument in Section 3.1 is not even really about model editing. It's about LLMs and/or LLM APIs.
>
> The points we make in 3.1 are about considering LLMs as "fact repositories" - which they are not. It is not about LLMs in general - but about approaches that assume LLMs "contain facts" and build from there. Section 3.1 challenges the presupposition underlying model editing.
>
> > For Section 3.2, the same argument could be used to argue against LLMs --
> > and even against the whole idea of indexing the Web and using it to learn things about the world!
>
> The arguments in 3.2 do not pertain to LLMs as artifacts that encapsulate "knowledge of language" - but instead as LLMs perceived as "knowledge of the world".
>
> There is a fundamental difference between Web indexes and LLMs: Web Indexes contain a repository of identifiable resources.
> They return identified links to resources. If you remove or blacklist a resource, it will not be returned by the search engine.
> When you add a new resource, it can be returned.
> LLMs once they are trained explicitly lose the coupling between sources and generations through generalization.
>
> > Search engines are vexing for the paper's arguments concerning "factuality". All the concerns raised about factuality hold for traditional search, including the arguments in 3.1. If lots of people on the Web say that Los Angeles is the capital of California, then n-gram-based search will also surface those results. The issue here is conflating factuality with grounding in a database. AI people should never promise factuality, but they should be able to deliver grounding or provenance.
>
> Regardless of the method used by search engines, the results returned by a search engine include links to the source resources.
> If the link refers to https://qanon - it is now the responsibility of the reader to give it credence (or not).
> We do agree on the basic desirable property - *grounding or provenance* - we take it for granted that search engines do provide provenance - LLMs do not.
>
> > The argument in 3.2 (and the metaphor of "emptying the ocean with a spoon") needs to be further substantiated. After all, changing a single node in a knowledge graph can have wide-ranging effects that spread automatically through the network. For example, changing the name of the U.S. president could change the name of the dog owned by the president's mother's first cousin, assuming the knowledge graph has the right edges and so forth. If we could train LLMs to have this kind of structure, then we would be able to change a lot of things with a small database of facts. But, anyway, all of this needs to be properly benchmarked against what we have now with Web search, Wikipedia editing, etc.
>
> Indeed - changing facts in knowledge bases is extremely complex. It used to be the topic of active research in Truth Maintenance Systems and was shown to be an NP-complete hard computational problem (Vladislav Rutenburg. *Complexity classification of truth maintenance systems*. In STACS, pages 372 – 382, 1991). There is no a priori reason to believe LLMs provide a foundation to achieve this goal.  Yet, model editing methods are proposed when as you state, it is quite a problem to even define a proper benchmark to measure these effects in a comprehensive manner.
>
> > I appreciate that section 4, on alternative approaches, notes that these alternatives suffer from all of the major problems that the paper identifies with model editing. But this is also a signal that the paper has set its sights too high.
>
> The alternatives we survey explicitly **reject** the assumption that LLMs are "facts repositories" - this is what makes them "better alternatives" in our analysis.  Yet while retrieval-based models seem to simply avoid the problem altogether (by separating the "facts repository" from the LLM), they do not completely avoid it for the reasons we enumerate (mainly because of the risk introduced by the "composition" component) - and thus, the concerns we enumerate deserve further attention.
>
> In contrast, concept erasure does not suffer from the risks we identify.

---

### Official Review · Reviewer_Y4br · 2023-08-09

**Soundness:** 3

**Excitement:**

3: Ambivalent: It has merits (e.g., it reports state-of-the-art results, the idea is nice), but there are key weaknesses (e.g., it describes incremental work), and it can significantly benefit from another round of revision. However, I won't object to accepting it if my co-reviewers champion it.

**Paper Topic And Main Contributions:**

In this position paper, the authors argue that pursuing model editing-based methods for improve the factuality of (large) language models is not a hopeful path, with the following arguments:

1) The probabilistic nature and training objectives of LMs are inherently unsuitable for factual purposes. For example, LMs are in some sense expected to imitate frequent falsehoods in the training data, and there is no built-in consistency or robustness for LM generation.

2) There are tremendously many facts being updated everyday, and editing model knowledge for all of them is not practical. It's also inevitable that bias would be introduced if we selectively choose which facts to edit.

3) Extensive model editing could lead to catastrophic forgetting and other performance downgrades, as found empirically in recent work.

Based on these, the authors suggest that alternative approaches such as retrieval-augmented/attributed generation and concept erasure are more promising.



**Reasons To Accept:**

The arguments in this paper are overall interesting and quite convincing, with adequate supporting evidence from research studies and other sources. The topic of factuality in LMs is also of central importance in NLP, especially with the recent rapid developments of large LMs and the new trend of LM-based search engines.

**Reasons To Reject:**

I find the discussion around alternatives to model editing a bit unsatisfying overall - while the content does cover in a rather comprehensive way the possible alternative paths towards factuality in LMs (e.g., retrieval, grounding, continual training, concept erasure, etc.), the discussion is overall closer to a literature survey and I don't find much critical opinions on the way forward for these alternative directions and approaches of resolving the challenges in these directions. One example is the issue regarding the difficulty of attributing generations to internal parameters/external stores as discussed in line 284-304, which is arguably a problem of central importance of this topic. In the end, the authors don't seem to lay out a solution forward, and it seems that these alternative paths face challenges somewhat similar to those in model editing. I would expect to see more fruitful and critical thoughts & discussions around these key topics, which I believe should be present in a position paper.

**Reproducibility:**

N/A: Doesn't apply, since the paper does not include empirical results.

**Reviewer Confidence:**

4: Quite sure. I tried to check the important points carefully. It's unlikely, though conceivable, that I missed something that should affect my ratings.

---

> ### Author Rebuttal · Authors · 2023-08-24
>
> ### Reaction to Reasons To Reject
>
> > I find the discussion around alternatives to model editing a bit unsatisfying overall:
> > while the content does cover in a rather comprehensive way the possible alternative paths towards factuality in LMs (e.g., retrieval, grounding, continual training, concept erasure, etc.), the discussion is overall closer to a literature survey and I don't find much critical opinions on the way forward for these alternative directions and approaches of resolving the challenges in these directions.
>
> The main point of our position paper is a critique of the "model editing" strategy together with its underlying assumption that an LLM
> can be considered as a knowledge base or "fact repository". The alternatives we survey explicitly **reject** this assumption that LLMs are "fact repositories".
>
> We feel that we do provide a critical assessment of the proposed alternatives - mainly: retrieval-based models (284-317), concept erasure (343-351).
>
> While retrieval-based models seem to simply avoid the problem altogether (by separating the "fact repository" from the LLM), they do not completely avoid it for the reasons we enumerate - and thus, the concerns we enumerate deserve further attention.
>
> In contrast, concept erasure does not suffer from the risks we identify.
>
> > One example is the issue regarding the difficulty of attributing generations to internal parameters/external stores as discussed in line 284-304, which is arguably a problem of central importance of this topic. In the end, the authors don't seem to lay out a solution forward, and it seems that these alternative paths face challenges somewhat similar to those in model editing.
>
> Indeed, this is a position paper. We aim at raising researchers' awareness of the risks associated with specific research objectives.
> We do recommend to aim for solutions that do not consider LLMs to be fact repositories, and instead to make efforts to attribute generations to identifiable sources - which remains an open research objective, but one that does not intrinsically rely on the problematic assumption that LLMs are fact repositories.

---

### Official Review · Reviewer_8HBp · 2023-08-12

**Soundness:** 4

**Excitement:**

4: Strong: This paper deepens the understanding of some phenomenon or lowers the barriers to an existing research direction.

**Paper Topic And Main Contributions:**

The paper addresses the important topic of model editing, which aims to update an existing model to elicit a desired outcome. The paper discusses eloquently of the drawback and systemic issues in the existing formulation of the model editing problem and how it violates certain properties necessary for trust in such systems. The paper also discusses interesting research directions of slightly different formulations aimed at grounding LLM generated outputs. The paper offers interesting arguments and perspective overall.

**Reasons To Accept:**

The paper provides a very strong case against existing model editing formulation. A nice survey of method have been presented regarding existing model editing methods and problems associated with them such as fact popularity bias, catastrophic forgetting and robustness to changes.

The paper also briefly discusses alternate approaches aimed at mitigating the hallucination issue by decoupling the factual memory component from LLM by relying on external knowledge stores, factual knowledge bases and posthoc editing of model parameters.

The objective mismatch between model pretraining and desired control at inference is very well argued with suitable citations

**Reasons To Reject:**

No major weaknesses in the position paper however, though not mandatory it would have been nice to provide some more grounding for the arguments and ciatations against implausibility of model editing.

More theoretically grounded works that show that model editing is not posssible (like unlearning has been shown to be theoretically hard which involves model editing) and also how continual learning is hard (https://arxiv.org/abs/2006.05188) would further support the arguments better.

More recent citations on grounding LLM outputs might be relevant like:

https://arxiv.org/abs/2302.12813

https://arxiv.org/pdf/2302.02662.pdf

**Reproducibility:**

N/A: Doesn't apply, since the paper does not include empirical results.

**Reviewer Confidence:**

5: Positive that my evaluation is correct. I read the paper very carefully and I am very familiar with related work.

---

> ### Author Rebuttal · Authors · 2023-08-24
>
> Thank you for the positive review!
>
> We will definitely consider the relevant links you have provided as well as classical complexity results from Assumption-based Truth Maintenance Systems (ATMS are NP-complete - Vladislav Rutenburg. Complexity classification of truth
> maintenance systems. In STACS, pages 372 – 382, 1991).

---

### Meta-Review · Area_Chair_nwuK · 2023-09-18

**Recommendation:** 4

**Metareview:**

The authors write a convincing position paper against utilization of editing techniques to modify model output to be more desirable. Posthoc approaches can have unchecked effects on other axes of model knowledge and generation, and requires more reflections, as identified by authors. While reviewers point out that the alternate approaches to editing that authors contribute are not substantial, I don’t think that is needed or the central point of the paper. As a lot of work quickly evolves to do more ‘editing’, this work comes as a very timely counterpoint and will benefit the community.
I would encourage the authors to reflect further on the alternative they propose as well as the need of ‘factual checking’ to contextualize their position.

---

### Decision · Program_Chairs · 2023-10-07

**Decision:**

Accept-Findings

**Comment:**

The authors write a convincing position paper against utilization of editing techniques to modify model output to be more desirable. Posthoc approaches can have unchecked effects on other axes of model knowledge and generation, and requires more reflections, as identified by authors. While reviewers point out that the alternate approaches to editing that authors contribute are not substantial, I don’t think that is needed or the central point of the paper. As a lot of work quickly evolves to do more ‘editing’, this work comes as a very timely counterpoint and will benefit the community.
I would encourage the authors to reflect further on the alternative they propose as well as the need of ‘factual checking’ to contextualize their position.